# Associations of hurricane exposure and forecasting with impaired birth outcomes

Jacob Hochard [1] ✉, Yuanhao Li[2] & Nino Abashidze[1]

Early forecasts give people in a storm's path time to prepare. Less is known about the cost to society when forecasts are incorrect. In this observational study, we examine over 700,000 births in the path of Hurricane Irene and find exposure was associated with impaired birth outcomes. Additional warning time was associated with decreased preterm birth rates for women who experienced intense storm exposures documenting a benefit of avoiding a type II forecasting error. A larger share of this at-risk population experienced a type I forecasting error where severe physical storm impacts were anticipated but not experienced. Disaster anticipation disrupted healthcare services by delaying and canceling prenatal care, which may contribute to storm-impacted birth outcomes. Recognizing storm damages depend on human responses to predicted storm paths is critical to supporting the next generation's developmental potential with judicious forecasts that ensure public warning systems mitigate rather than exacerbate climate damages.

The best available climate science predicts an increase in high-intensity[1] and less predictable[2,3] tropical storms. Despite the physical damages from these storms totaling over 500 billion dollars since 2004, National Oceanic and Atmospheric Administration (NOAA) National Hurricane Center (NHC) funding allocated to forecasting tropical storm threats continues to diminish[4]. Media coverage supports advanced warning systems by forecasting potential threats to the masses. The goal of disaster forecasts is to avert damages to infrastructure, human health and well-being while recognizing that the broadcast itself will increase psychological stress in viewing populations[5]. Yet, no large-scale study exists documenting the relationship between forecast accuracy and human health impacts in hurricane-threatened populations.

The release of highly uncertain disaster forecasts may represent a public health threat for several reasons. Disaster-related media coverage has long been shown to contribute to posttraumatic stress disorder symptoms in viewers[6,7]. New evidence reveals that forecasted posttraumatic stress symptoms, leading up to a hurricane event, influences the public's mental health before and after a hurricane event[5]. Taken together, NHC storm forecasts, such as the "Cone of Uncertainty," which are often misinterpreted by the public[8] but widely disseminated by the media, may cause substantial distress to viewing populations. Such a public health threat is most preventable in communities that expect and prepare for a hurricane exposure that does not end up generating physical impacts, i.e., a type I forecasting error.

Experiencing a disaster during pregnancy can impair birth outcomes[9–12] and disrupt access to healthcare services[13–16], which may have long-run implications for the unborn child's livelihood[11,17,18]. In utero exposure to stress[12], environmental toxins[19,20] and disrupted access to health services[21] are leading explanations for observed reductions in birth weight and gestation lengths. In the context of tropical storms, in utero exposures have also led to abnormal conditions of the newborn baby (e.g., ventilator dependence and meconium aspiration syndrome)[22]. Causal linkages between these birth outcomes and later life disease prevalence[23,24], mental health[12], aptitude, educational attainment and future wages[17] have been established. No study to date has isolated these underlying mechanisms empirically and measured the extent to which institutions influence birth outcomes by disseminating uncertain disaster forecasts to the public.

The purpose of this study is twofold. First, we investigate empirically the impact of in utero exposure to Hurricane Irene on a variety of birth outcomes, including birth weight, gestation length and incidence of low birth weight and preterm birth outcomes. The sample focuses on more than 700,000 births in representative communities

[1]Haub School of Environment and Natural Resources, University of Wyoming, Bim Kendall House, 804 E Fremont Street, Laramie, WY 82072, USA. [2]SNF – Centre for Applied Research at NHH, Helleveien 30, N-5045, Bergen, Norway. ✉e-mail: JHochard@uwyo.edu

across North Carolina (Supplementary Fig. 1). Consistent with previous disaster and stressful events literature, we hypothesize that the hurricane exposure will reduce birth weights and gestation periods leadings to more frequent low birth weight and preterm birth outcomes. Examination of Hurricane Irene as a natural experiment is conducted where birth outcomes leading up to the storm's landfall serve as a baseline of comparison against births within the same zip code that occurred after the storm and experienced in utero exposures. Second, we investigate potential mechanisms underlying observed birth impacts with the expectation that groundwater contamination and intensive rainfall and wind would be an important contributor to measured birth outcome effects.

## Results

We reported birth impacts as an average across all rainfall intensity bands with associated 95% confidence intervals (CI) in parentheses. The estimated birth impacts were overlayed with a cumulative distribution of wind speed exposures in order to visually explore the association between estimated birth outcomes and wind intensity (Fig. 1). The average in utero exposure to Hurricane Irene was associated with reduced birth weights by 12.7 g (5.4–20.0 g), which represented a 0.17–0.61% reduction on the birth weight sample mean ($\bar{x}^{bw} = 3263.7$ g). The largest treatment effect of 14.4 g (−3.9 to −25.0 g) was estimated for populations receiving hurricane-force winds and a one-day maximum rainfall in excess of 10 inches and the smallest

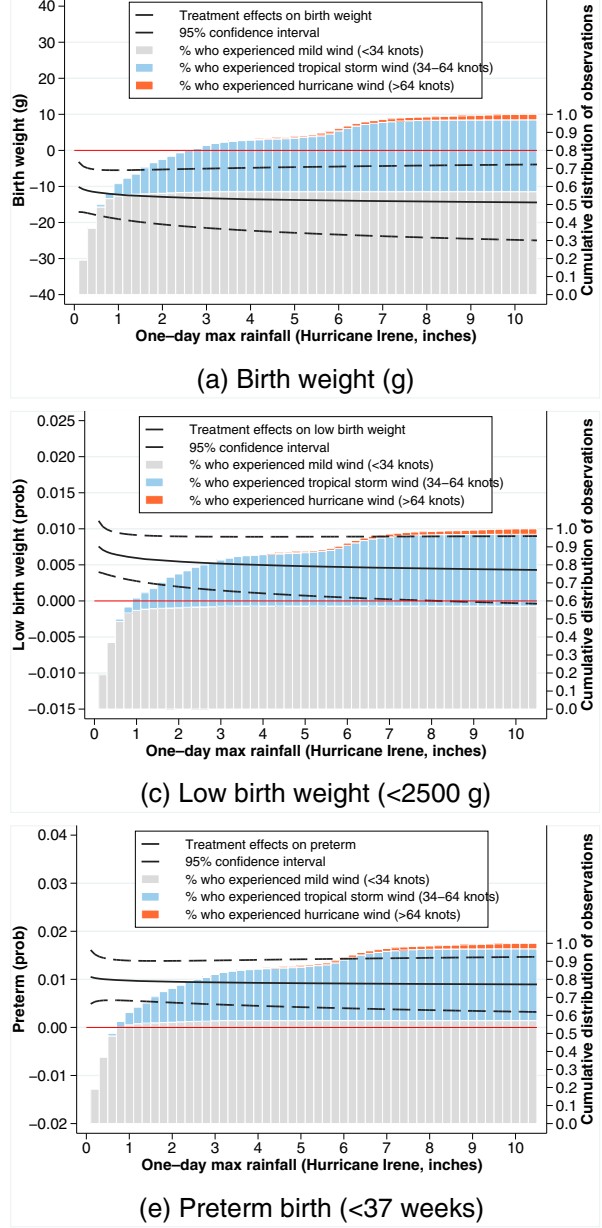

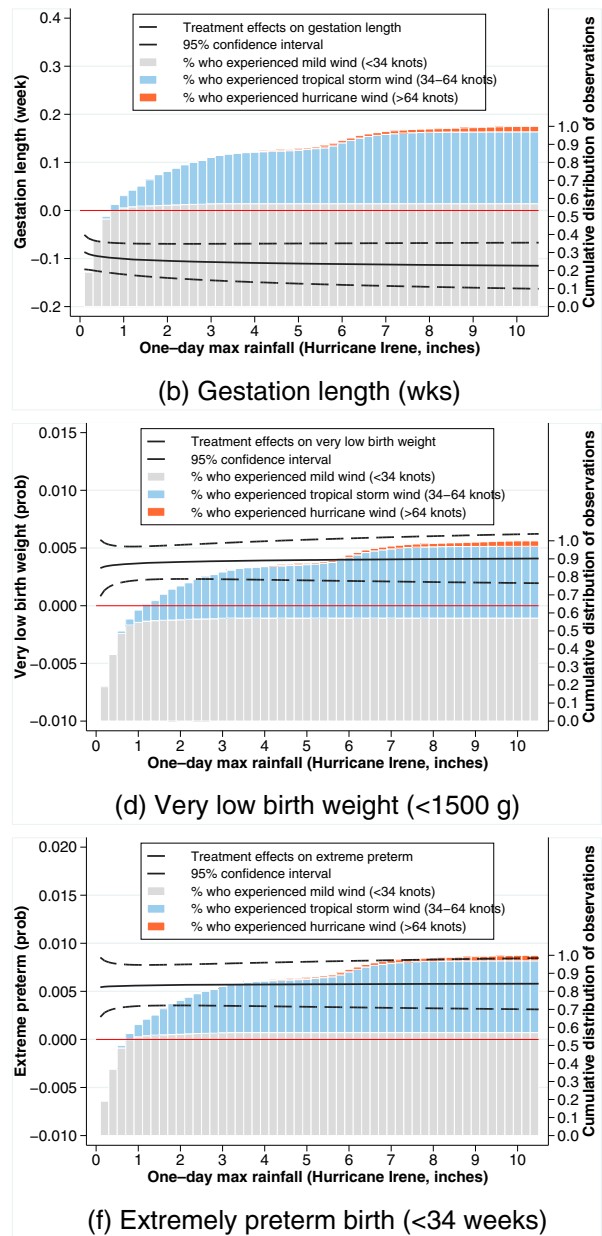

**Fig. 1 | Estimated treatment effect of Hurricane Irene exposure on birth outcomes plotted across rainfall intensity at pregnant women's residential addresses.** The treatment effects measured for a variety of birth outcomes including: **a** birth weight (g), **b** gestation length (weeks), **c** binary (0,1) incidence of low birth weight <2500 g outcomes, **d** binary (0,1) incidence of very low birth weight <1500 g outcomes, **e** preterm birth (<37 weeks), and **f** extremely preterm birth (<34 weeks). Treatment effects were estimated against a baseline of births from the same zip code that occurred with expected delivery dates within the 5

years leading up to the Hurricane Irene's disaster declaration date of August 25, 2011. Rainfall at residential address was used as an indicator of exposure intensity represented by the 1-day maximum rainfall from August 14, 2011 to September 4, 2011, which encompassed the hurricane event's impact on North Carolina. Estimated treatment effects were overlayed with a cumulative distribution of wind speed exposures. Rainfall source: Parameter-elevation Regressions on Independent Slopes Model (PRISM) climate group Time Series Values for Individual Locations. Source data are provided as a Source Data file.

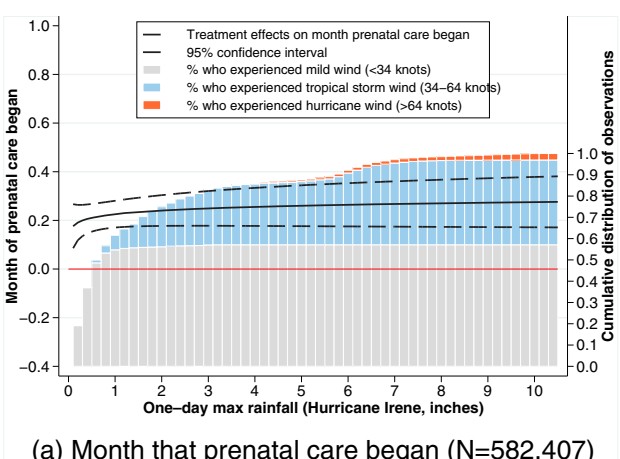

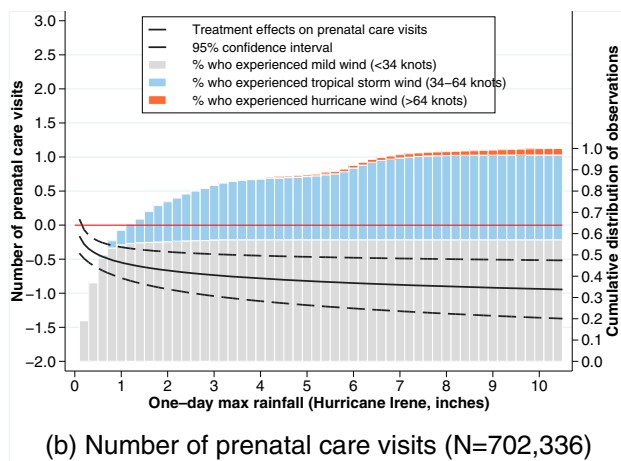

| (a) Month that prenatal care began (N=582,407) | (b) Number of prenatal care visits (N=702,336) |

**Fig. 2 | Estimated treatment effect of Hurricane Irene exposure on the month that prenatal care began and the number of prenatal care visits.** The treatment effects measured for prenatal care indicators including: **a** The month that prenatal care began and **b** the total number of prenatal care visits. Rainfall at residential address was used as an indicator of exposure intensity represented by the one-day maximum rainfall from August 14, 2011 to September 4, 2011, which encompassed the hurricane event's impact on North Carolina. Estimated treatment effects were overlayed with a cumulative distribution of wind speed exposures. Rainfall source: Parameter-elevation Regressions on Independent Slopes Model (PRISM) climate group Time Series Values for Individual Locations. Source data are provided as a Source Data file.

treatment effect of 10.1 g (−3.2 and −17.1 g) was estimated for populations receiving less than 1 inch of rainfall and only mild winds (Fig. 1a).

Gestation lengths were also shortened following in utero exposure by an average of 0.10 weeks (0.07–0.14 weeks), which represented a 0.18–0.36% reduction on the gestation length sample mean ($\bar{x}^{gest} = 38.5$ weeks). The largest treatment effect of 0.11 weeks (0.07–0.16 weeks) was estimated for populations receiving hurricane-force winds and a 1-day maximum rainfall in excess of 10 inches and the smallest treatment effect of 0.09 weeks (0.05–0.12 weeks) was estimated for populations receiving less than 1 inch of rainfall and only mild winds (Fig. 1b).

A similar pattern was revealed for increased likelihood of experiencing low birth weight, very low birth weight, preterm and extremely preterm birth outcomes following in utero exposure to Hurricane Irene. Each of these birth impacts was statistically significant in difference from zero but the magnitudes of these effects were statistically indistinguishable from each other across our wind and rainfall indicators for exposure intensity (Fig. 1). The incidence of low birth weight outcomes increased by 0.56 percentage points (0.22–0.90 percentage points), which represented a 2.52–10.34% increase in the likelihood of a low birth weight outcome on the sample mean ($\bar{x}^{lbw} = 0.087$) (Fig. 1c). The incidence of very low birth weight outcomes increased by 0.38 percentage points (0.23–0.52 percentage points), which represented a 15.33–34.67% increase in the likelihood of a very low birth weight outcome on the sample mean ($\bar{x}^{vlbw} = 0.015$) (Fig. 1d). The incidence of preterm births increased by 0.96 percentage points (0.53–1.38 percentage points), which represented a 5.20–13.53% increase in the likelihood of a preterm birth on the sample mean ($\bar{x}^{pre} = 0.102$) (Fig. 1e). The incidence of extremely preterm births increased by 0.56 percentage points (0.35–0.78 percentage points), which represented a 12.07–26.90% increase in the likelihood of an extremely preterm birth on the sample mean ($\bar{x}^{expre} = 0.029$) (Fig. 1f).

We further investigated the nature of potential physical exposures by focusing on statewide groundwater contamination. We examined over 17,000 private well water samples that were collected by county health offices statewide and processed through the North Carolina State Laboratory of Public Health. We focused on nitrate, manganese, lead, chromium, cadmium and arsenic results, which are all known to disrupt in utero development and have exposure pathways related to major storm events[25–30]. While these samples were not taken from the same residences of our pregnant women sample, both data sets had

statewide and residence-level coverage and used the same selection into treatment window around Hurricane Irene. We found no meaningful relationship between storm exposure intensity and private well water contamination rates at the Environment Protection Agency's recommended maximum contaminant load level (Supplementary Fig. 2) or the North Carolina State Laboratory's minimum detection limit (Supplementary Fig. 3). This null finding did not rule out contaminated groundwater exposures from Hurricane Irene and is not the primary purpose of our contribution. However, if these contamination events were the driving force behind our results, we might have expected such evidence to appear in this sample.

We also examined a suite of medical risk factors reported for each pregnant woman within our data set (Supplementary Fig. 4). We focused on the incidence of prepregnancy hypertension and having previously had a poor pregnancy outcome for each individual within our sample. Previously poor pregnancies included those that resulted in perinatal death, small-for-gestational age or intrauterine growth restricted births. We found no relationship between selection into treatment (or intensity of treatment) and the presence of a prepregnancy hypertension or previous poor pregnancy diagnosis (Supplementary Fig. 4). We then focused on the incidence of gestational hypertension and eclampsia that may had developed during pregnancy. To the extent that maternal stress from experiencing a severe storm event was driving observed birth impacts, we might expect gestational hypertension and eclampsia rates to be elevated in our treatment group. We found no evidence of increased incidence of gestational hypertension or eclampsia relative to our baseline group (Supplementary Fig. 4).

We turned our attention to the disruption of healthcare services from hurricane exposure that might impact birth outcomes (Fig. 2). For each birth in our analysis, we examined the impact of hurricane exposure intensity on the month that prenatal care began following the clinically determined conception date ($N = 582,407$) and the total number of prenatal care visits that occurred throughout the pregnancy ($N = 702,336$). The average in utero exposure to Hurricane Irene was associated with a delayed first prenatal care appointment by 0.24 months (0.18–0.30 months), which represented an approximate 1 week or 6.92–11.54% delay on the sample mean of when prenatal care was initiated ($\bar{x}^{beg.} = 2.60$) (Fig. 2a). The total number of prenatal care appointments was reduced on average by 0.63 appointments (0.37–0.89 appointments) following in utero exposure to Hurricane

**Table 1 | Treatment effect of an additional 6-h "Cone of Uncertainty" advisory**

| Birth outcomes | Effect of an additional advisory | | |
|---|---|---|---|
| | Rain > 2 in | 1 < Rain ≤ 2 in | Rain ≤ 1 in |
| Birth weight (g) | 1.391 | –4.815 | –4.111*** |
| | (1.099) | (3.023) | (1.449) |
| Mean dept. var. | 3252.378 | 3281.767 | 3263.981 |
| Gestation length (weeks) | 0.00357 | –0.0201 | –0.0201 |
| | (0.00304) | (0.0123) | (0.0131) |
| Mean dept. var. | 38.526 | 38.544 | 38.543 |
| Low birth weight (<2500 g) | –0.000629 | 0.00122 | 0.00206* |
| | (0.000403) | (0.00113) | (0.00105) |
| Mean dept. var. | 0.090 | 0.084 | 0.087 |
| Very low birth weight (<1500 g) | 0.000131 | 0.000488 | 0.000846* |
| | (0.000218) | (0.000579) | (0.000484) |
| Mean dept. var. | 0.016 | 0.015 | 0.015 |
| Preterm (<37 weeks) | –0.00126** | 0.00197 | 0.00114* |
| | (0.000496) | (0.00152) | (0.000666) |
| Mean dept. var. | 0.103 | 0.102 | 0.102 |
| Extreme preterm (<34 weeks) | 0.000273 | 0.000514 | 0.00139* |
| | (0.000290) | (0.000898) | (0.000703) |
| Mean dept. var. | 0.030 | 0.030 | 0.029 |
| Observations | 177,425 | 95,056 | 436,233 |

All models include month of birth and zip code fixed effects. The estimates for binary outcome variables are based on a linear probability model. Standard errors clustered at the county level are in parentheses. All *t*-tests are two sided (not reported). All econometric analyses were conducted using Stata/MP 16.1. Source data are provided as a Source Data file.
***$P < 0.01$, **$P < 0.05$, and *$P < 0.1$.

Irene, which represented a 3.03–7.29% reduction on the sample mean of total prenatal care appointments ($\bar{x}^{app.} = 12.21$) (Fig. 2b). Similar to the observed impacts of Hurricane Irene on birth outcomes, we observed that prenatal care disruptions were significant but varied little across the intensity of storm exposures.

The average individual that received heavy rainfall experienced an average of 15.9 6-h periods within Hurricane Irene's cone of uncertainty (approximately 95 h). For these individuals, the prediction of direct hurricane exposure was ex post correct and additional time spent within the cone of uncertainty served as an accurate risk signal to prepare for imminent exposure. We found that the marginal effect of an additional 6-h window of preparation for these heavily exposed populations had no meaningful impact on birth outcomes, gestation length or the incidence of low birth weight, very low birth weight or extreme preterm birth outcomes (Table 1, column: Rain > 2 in). We found that additional advisories for this group of women was associated with a statistically significant decrease in the likelihood of having a preterm birth, which represented a 1.2% reduction on the heavily exposed sample mean ($\bar{x}^{pret} = 0.103$) (Table 1, column: Rain > 2 in).

The average individual that received light rainfall experienced an average of 6.5 6-h periods within Hurricane Irene's cone of uncertainty (approximately 39 h). For these individuals, the prediction of direct hurricane exposure was ex post incorrect and additional time spent within the cone of uncertainty served as an inaccurate risk signal that may have disrupted unnecessarily planned healthcare services. We found that the marginal effect of residing within the cone for an additional 6-h window was associated with decreased birth weights by 4.1 g for this lightly exposed population, which represented a 0.13% reduction in birth weight on the lightly exposed sample mean

($\bar{x}^{bw} = 3264.1$) (Table 1, column: Rain ≤ 1 in). For this group, we also found that an extended time of anticipating direct impact was associated with a marginally significant increase in the incidence of low birth weight, very low birth weight, preterm and extreme preterm births (Table 1, column: Rain ≤ 1 in). The marginal impact on low birth weight incidence was 0.0021, which represented a 2.4% increase in the likelihood of a low birth weight outcome on the lightly exposed sample mean ($\bar{x}^{lbw} = 0.0869$). The impact on very low birth weight outcomes was relatively larger, 0.0008, which represented a 5.4% increase in the likelihood of a very low birth weight outcome on the lightly exposed sample mean ($\bar{x}^{vlbw} = 0.0147$). We observed a similar trend for preterm and extreme preterm births. The marginal impact on preterm incidence was 0.0011, which represented a 1.1% increase in the likelihood of a preterm birth on the lightly exposed sample mean ($\bar{x}^{pret} = 0.1016$). The impact on extreme preterm births was relatively larger, 0.0014, which represented a 4.9% increase in the likelihood of a very low birth weight outcome on the lightly exposed sample mean ($\bar{x}^{expret} = 0.0287$).

## Discussion

We observed evidence that in utero exposure to Hurricane Irene created widespread and detrimental impacts to birth outcomes. Across all rainfall intensity bands analyzed in the paper, we detected consistent birth effects that were statistically distinguishable from zero. Although we would expect that a higher intensity of rainfall and wind would be associated with more drastic birth impacts, the magnitudes of these estimated effects did not increase with storm exposure intensity. The consistency in our measured birth impacts across storm exposure intensity was unexpected and may suggest that birth impacts were being driven by a mechanism other than physical storm exposures (e.g., rainfall, wind and groundwater contamination in flooded areas).

We might expect that statewide groundwater contamination would explain the observed birth impacts. During Hurricane Irene, over 2 million individuals that represented over 20% of North Carolina's population[31] relied on private wells that were federally unregulated and particularly vulnerable to contamination from severe weather and flooding events[32–35]. We found no evidence that birth impacts were driven by direct physical impacts of Hurricane Irene (e.g., high winds or flooding) or indirect physical exposures (e.g., groundwater contamination that resulted in flooded areas).

We further found no evidence that the observed birth impacts were explained by a systematic geographic sorting along socioeconomic lines, which may have occurred during our sample window. In such a case, we would expect prediagnosed medical risk factors to vary systematically between our treatment and control groups. Importantly, we observed evidence that hurricane exposure created prenatal care disruptions that varied little across the intensity of storm exposures. Such a finding suggests that the anticipation of hurricane exposures and associated institutional responses to that anticipation, rather than the physical direct and indirect impacts from the storm itself, may be a key factor.

The evidence presented in this work is consistent with the notion that in utero tropical storm exposures create abnormal birth conditions[22]. Consistent with the broader disasters literature[9–12], we build on[22] with clear evidence that storm exposures reduced birth weights and gestation lengths while increasing the likelihood of preterm and low birth weight outcomes. We also provide the first evidence that uncertain hurricane forecasts lead to individual-level disruptions in healthcare services. These impacts on birth outcomes are similar in magnitude to those found in response to other traumatic events experienced during pregnancy, such as nearby terrorist attacks[36], bereavement[12] and financial hardship[37]. A key distinction is that the driving mechanism of exposure is a public warning system that is designed to mitigate rather than exacerbate the impacts of storm events on threatened populations. Studies such as ours are a first step to timing the optimal dissemination of disaster forecasts.

The presented findings are clinically relevant in addition to being statistically robust. Relying on British National Child Development Survey data[38], show that low birth weight children weighing less than 2500 g were more than 25% less likely to pass high school English and math exit examinations and were also less likely to be employed at the age of 33. In our work, we show that in utero exposure to Hurricane Irene created a 2.52–10.34% increase in the likelihood of crossing this critical low birth weight threshold, which is increased further by storm exposure anticipation. For our birth weight outcomes, we find general effects that range from 0.17 to 0.61% and increase similarly with storm exposure anticipation. For the average individual in our lightly exposed sample (Rain ≤ 1 in), direct effects of storm exposure and indirect effects from storm forecast-driven anticipation (on average 6.5 6-h windows within the Cone of Uncertainty), cumulative birth weight reductions are approximately 1–2%. The clinical impacts of these birth weight reductions are uncertain. However, the measured magnitude is well below the commonly cited "10% change" where disruption to later life outcomes, such as high school graduation rates, IQ, income and height have been documented[39].

Findings highlight the importance of understanding risk preferences of disaster-threatened populations and institutions. In the case of Hurricane Irene, the early release of the storm track forecast is likely to have triggered a precautionary response by patients and healthcare providers. The apparent decision to cancel healthcare appointments may be driven by risk averse preferences among these groups. However, the spatial extent of those cancellations was determined by the amount of forecast uncertainty. As such, it appears that this combination of risk averse preferences and forecast uncertainty during Hurricane Irene may have disproportionately harmed the unborn. On the margin, delaying the release of Hurricane Irene's storm forecast release may have improved birth outcomes (birth weight, low birth weight and preterm outcomes) for 2.5 women relative to each women for which the delay would have impaired birth outcomes (increased preterm births). Evaluating the impact of storm forecast uncertainty in this way has the potential to guide cost-benefit analyses for the research and development of improved storm prediction models. Before such approaches could be used in a useful way, further study is needed to understand how disaster-threatened populations might respond to delayed but higher accuracy forecasts.

The findings presented herein provide direction for several areas of future research. Empirically, heterogeneity in the estimated treatment effects should be explored to support future policy implications. Hurricane "experience" may mediate observed healthcare disruptions, which could be investigated by linking residential addresses with historical storm events and real estate records. Trimester of exposure should also be investigated to identify populations that are most vulnerable to disruptions in healthcare services and to formally validate whether prenatal care disruptions were the force driving observed birth outcomes. Behaviorally, our analysis is unable to predict the psychological impacts of a delayed storm forecast. The ambiguity (rather than uncertainty) surrounding a low-information scenario may trigger similar precautionary responses by individuals and institutions during the anticipation phase of delayed official storm forecasts. Here, future research should examine how public risk responses are likely to differ in disaster scenarios characterized by extreme ambiguity and the extent to which birth impacts from storm events are driven by physiological stress channels compared to institutional responses to situational stress. Although the analyzed data set includes all officially recorded births in North Carolina for the period 2006–2012, the study focuses on the impacts of Hurricane Irene, which may not be generalizable to other storm events that were more or less predictable and had different compositions of physical exposures (e.g., wind versus rainfall intensity leading to a variable propensity for flooding across watersheds). Better understanding these mechanisms that underlie population responses and institutional responses to disasters is essential to guiding future policies that might affect disaster preparedness.

## Methods

This study complies with ethical regulations for research on human subjects. The study is governed by East Carolina University's IRB 17-000354 and the written informed consents for all the data were waived by the IRB. Our analysis was based on the North Carolina Department of Health and Human Services (NCDHHS) vital statistics data set for all North Carolina live and still births from August 26, 2006 to June 14, 2012. The study followed Strengthening the Reporting of Observational Studies in Epidemiology (STROBE) reporting guideline for reporting observational studies. We constructed a data set of 710,186 North Carolina birth outcomes with associated prenatal care and medical risk factor information that were georeferenced at the residential address level. Birth outcomes included birth weight (g) and gestation length (weeks) variables that were used to create binary indicators for low birth weight (<2500 g), very low birth weight (<1500 g), preterm (<37 weeks) and extreme preterm (<34 weeks) outcomes. Associated prenatal care indicators included the number of prenatal care visits and the gestational month in which prenatal care began. Medical risk factors included indicators for prepregnancy and gestational hypertension.

We focused on the North Carolina impacts of Hurricane Irene, which made landfall August 27, 2011. Births that occurred in the 5 years prior to the hurricane's impact served as a baseline of comparison against births experiencing in utero exposure to Hurricane Irene. Incorporating zip code and monthly fixed effects ensured that our estimation procedure isolated the impact of hurricane exposure on birth outcomes rather than local, social, and institutional factors[40–43] and seasonal trends[44–47]. An annual time trend was also included to control for background trends in birth outcomes from 2006 to 2012. In all analyses, standard errors were clustered at the county level, which was the level of public health services and data collection throughout North Carolina. Clustering at the county level allowed for arbitrary serial correlation across births within the same county over time.

Pre-hurricane births served as a control group for post-hurricane births. Constructing the data set in this way exploited the fact that prenatal, but not postnatal, exposure to a disaster may influence birth outcomes. Our empirical strategy hinged on the assumption that pregnant women did not select into treatment. We presented layers of evidence that this assumption was reasonable and that our results gleaned insight into the causal nature of in utero exposure to hurricanes on birth outcomes. Precise birth dates and residential addresses allowed us to control for local neighborhood and seasonality effects that were known to otherwise influence birth outcomes. We then used variation in a woman's residential location relative to the NHC's ex ante "Cone of Uncertainty" forecasts and the hurricane's ex post storm track and associated rainfall intensities. We focused on the effects of in utero exposure to disaster stress by identifying women who were likely to anticipate direct hurricane impact but were not necessarily exposed to severe weather because of the storm's changing trajectory.

The selection of births included each woman's expected delivery date, which was defined as 280 days after the clinically estimated date of conception. An expected delivery date within 5 years leading up to the Hurricane Irene disaster declaration date, August 25, 2011, was placed into the control group and an expected birth date within the 280 days following Hurricane Irene was placed into the treatment group. Constructing the sample in this way included all births that experience prenatal or postnatal exposure to Hurricane Irene within the relevant time window.

Construction of the data set followed convention in the literature[12,48] and helped overcome two empirical challenges. First, opting to define the treatment window based on actual birth dates would have created a mechanical correlation between gestation length

and the likelihood that a pregnant woman experienced a hurricane, i.e., longer gestation lengths led to heavier birth weights and an increased likelihood that hurricane exposure occurred during the pregnancy. Second, a large literature and our findings suggested exposure to a disaster influences gestation length. Defining the treatment window based on expected birth dates, rather than actual birth dates, ensured that the treatment window was predetermined at the time of Hurricane Irene's arrival, i.e., there was no selection of women into treatment from exposure[12,48].

Formally, the sample selection contained a treatment group and a control group. The treatment group was all pregnant women residing in North Carolina during Hurricane Irene's disaster declaration date and within the first 40 weeks following their approximate date of conception (c). We defined the child's expected birth date as $e^b = c + 280$. The control group contained all women whose births were within $x$ days of Hurricane Irene. We included a full 5 years, $x = 1825$, of control group expected birth dates to ensure that we were able to account fully for seasonality effects in birth outcomes.

The sample selection of Hurricane Irene births followed[11] and[12] and was

$$S = \{i : \mathbf{1}[c \leq August\ 25, 2011 \leq b]_i = 1 | \mathbf{1}[b < August\ 25, 2011 \leq b + x]_i = 1\}.$$

Our data set included the residential address and birth date for each observation. We geocoded these residential addresses into $(x, y)$ decimal degree coordinate using the Aeronautical Reconnaissance Coverage Geographic Information System (ArcGIS) geocoder application programming interface (API) for Python (1.5.2). The coordinates were then converted into georeferenced points and used to calculate the distance of each residential location to the nearest point along a dissolved version of NOAA's preliminary best track from the National Hurricane Center's Geographic Information System (GIS) Archive – "Tropical Cyclone Best Track". Distance calculations were conducted using the UTM zone 17N projection. NOAA's NHC GIS sources were also used to overlay Hurricane Irene's "Cone of Uncertainty" predictions from advisory #7, which occurred on August 22, 2011 at 9:00 am and represented the first time that the Hurricane Irene 5-day cone approached the border of North Carolina, to advisory number #30A, which occurred on August 27, 2011 at 11:00 pm and represented the final intersection of North Carolina and Hurricane Irene's 5-day "Cone of Uncertainty" (Fig. 3).

We also merged the North Carolina State Laboratory of Public Health's statewide private drinking water well samples that were collected and processed during our Hurricane Irene treatment and control time periods. Comprehensive samples for arsenic, cadmium, chromium, lead, manganese and nitrate were all collected because they were known to cause adverse effects on birth outcomes when ingested during pregnancy. Furthermore, each contaminant could have been plausibly related to hurricane-exposure conditions or indicative of geographic sorting among pregnant women in response to risk. Together, these water samples helped us determine the underlying cause of observed birth outcomes and rule out the selection of women into treatment by geographic sorting along socioeconomic lines. Each inorganic analyte was coded as a binary outcome based on whether the sample exceeded the U.S. Environmental Protection Agency's safe drinking water standards. Our NCDHHS data set also included the number of prenatal visits that occurred during each woman's gestation period, the month that prenatal care began and information on whether prepregnancy hypertension, gestational hypertension, eclampsia and having prior had a poor pregnancy (including perinatal death and small-for-gestational age/intrauterine growth restricted births) were diagnosed as a medical risk factor.

Consistent with prior work[22], we hypothesized that hurricane exposures led to reduced birth weights and gestation lengths and an increased likelihood of a preterm and low birth weight outcomes.

Rainfall at each residential address was used as an indicator of exposure intensity represented by the one-day maximum rainfall from August 14, 2011 to September 4, 2011, which encompassed the hurricane event's impact on North Carolina. Rainfall data were from the Parameter-elevation Regressions on Independent Slopes Model (PRISM) climate group Time Series Values for Individual Locations. For the control group, rainfall data from Hurricane Irene were similarly overlaid with residential addresses as if these residences experienced physical exposures. However, our selection into treatment criteria ensured that only those women within our treatment group experienced uterine exposure whereas postnatal exposure in our control group cannot impact uterine conditions or birth outcomes of infants that were born before the hurricane's arrival. Conditioning on rainfall intensity in this way enabled a comparison of women in the treatment and control groups who presumably shared similar socioeconomic and demographic characteristics because they resided in the regions that would have been similarly exposed to Hurricane Irene's physical impacts.

To measure the impact of hurricane exposures, we compared birth outcomes among two groups of women who lived in the same zip codes and experienced antenatal or postnatal exposures. The comparison groups included a "treatment" group of exposed women whose births may have been affected through in utero exposures to the physical impacts of Hurricane Irene and a "control" group of exposed women whose birth outcomes predated the hurricane's arrival and could not have been impacted by its physical impacts. To refine our comparison groups, high-resolution rainfall intensities were predicted at each woman's residential address and implemented as our proxy exposure variable. While unlikely within any given zip code, if Hurricane Irene's physical exposures were systematically correlated with neighborhoods that were underserved in other ways that might impact birth outcomes (e.g., access to healthcare services or insurance), conditioning on Hurricane Irene's rainfall intensities ensured that our treatment and control groups mirrored one another. More specifically, we estimated the following equation:

$$
\begin{aligned}
y_{iymz} = \beta_0 + \beta_1 E_{iymz} + \beta_2 \ln R_{iymz} + \beta_3 E_{iymz} \times \ln R_{iymz} \\
+ \mu_m + Year + \zeta_z + \varepsilon_{iymz},
\end{aligned}
\tag{1}
$$

for a woman $i$ who resided in zip code $z$ whose birth took place in month $m$ of year $y$. The variable $E_{iymz}$ was a binary variable that takes the value of 1 if her birth was in the treatment group and 0 otherwise. That is, $E_{iymz} = \mathbf{1}[c \leq August\ 25, 2011 \leq e^b]_{iymz}$. The variable $\ln R_{iymz}$ was the natural logarithm of the 24-h maximum rainfall that the woman experienced during the hurricane week. The variables $\mu_m$ and $\zeta_z$ were month and zip code fixed effects. The variable $Year$ was a linear year trend. The dependent variable, $y_{iymz}$ was the birth outcome of each woman. Our estimating equation resembled[12] but included an exposure intensity variable that was interacted with the treatment group binary variable. Mediating the intensity of exposure in this way allowed us to estimate non-linear impacts of hurricane exposure, across rainfall intensity, on North Carolina's birth outcomes. Standard errors for all regressions were clustered across the State of North Carolina's 100 counties, which was the level of public health services and data collection throughout North Carolina. Clustering at the county level allowed for arbitrary serial correlation across births within the same county over time.

We first estimated the average treatment effect of hurricane exposure on birth outcomes and then calculated the predicted birth outcomes for each woman in our sample based on their actual exposure. More specifically, after Equation (1) was estimated, we calculated the predicted impact on an individual woman's birth outcome, $\psi$, as a

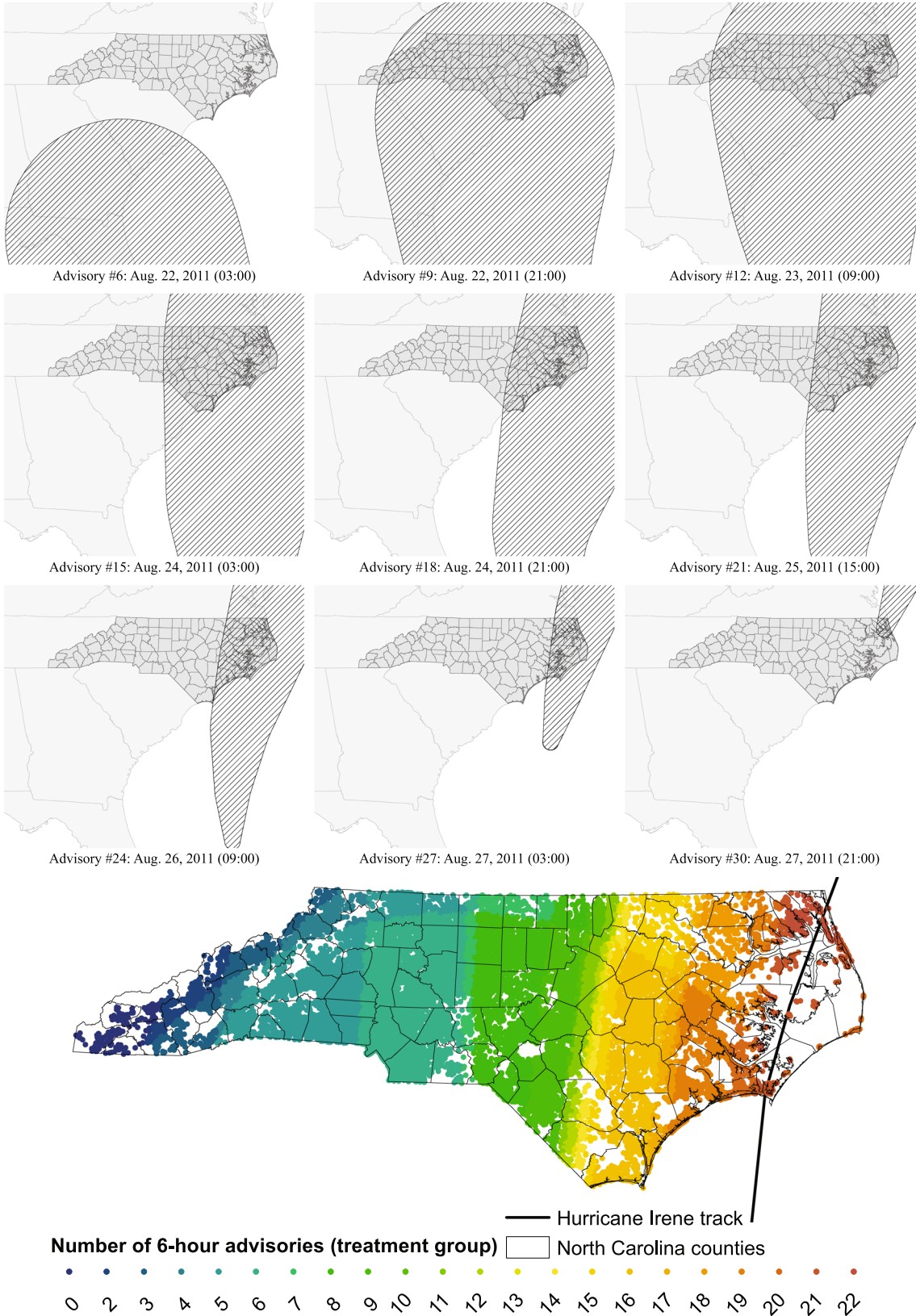

**Fig. 3 | Total number of 6-h advisories (a sample of which are shown here) experienced within the Cone of Uncertainty for the residential address of each woman within Hurricane Irene's treatment sample.** The underlying raw data are obtained from the National Oceanic and Atmospheric Administration (NOAA) National Hurricane Center (NHC). Source data are provided as a Source Data file.

function of $R_{iymz}$

$$\psi(R_{iymz}) = y|_{E=1} - y|_{E=0} = \beta_1 + \beta_3 \ln R_{iymz}, \qquad (2)$$

which was the basis for Figs. 1 and 2, Supplementary Figs. 2 and 4, and Table 1 with expanded summary statistics available in Supplementary Table 1.

To better understand the connection between hurricane anticipation and observed birth impacts, we overlaid the National Hurricane Center's "Cone of Uncertainty" forecasts for Hurricane Irene with the residential addresses of all pregnant women within our sample. The spatial extent of the Cone of Uncertainty represents a zone that will contain the "eye" of an impending hurricane with approximately 66% confidence. Variation in hurricane anticipation is given by the total hours that each residential address within our sample spends within this cone (Fig. 3). The cone first overlapped with North Carolina at 9:00 am on August 22, 2011 and scraped across the state until the hurricane's eye was over the northeastern corner of the state at 9:00 pm on August 27 (Fig. 3).

We stratified our sample into three categories that experienced light rainfall (<1 inch of rain within the most intensive 24-h period), moderate rainfall (1–2 inches of rain within the most intensive 24-h period) and heavy rainfall (>2 inches of rain within the most intensive 24-h period) during Hurricane Irene. Following the same empirical approach as previous analyses, our baseline group of comparison represented births from residential addresses that would have experienced the same light, moderate and heavy rainfall conditions if their pregnancies had overlapped with Hurricane Irene, i.e., births occurred in the same location but at a slightly later time. Separating the sample into categories of physical exposure allowed us to measure the additional benefit (or harm) that resulted from advanced warning that signaled potential impact ahead of a storm event that ended up bringing either mild, moderate or severe weather.

In other words, our approach isolated both (i) the reproductive health benefits of an advanced warning system that avoids a type II forecasting error—i.e., correctly provides additional warning time to vulnerable populations that received intense physical exposures—and the reproductive health harm of an advanced warning system that committed a type I forecasting error, i.e., incorrectly provided additional warning to vulnerable populations that only received mild physical exposures. In such a latter case, additional exposure anticipation may lead to the (ex post unnecessary) cancellation of prenatal care appointments, which may inadvertently cause harm to birth outcomes.

To examine the effect of hurricane anticipation (i.e., an additional 6-h window within Hurricane Irene's predicted "cone of uncertainty") on birth outcomes, we augmented the estimating equation as follows:

$$y_{iymz} = \beta_0 + \beta_1 E_{iymz} + \beta_2 \ln R_{iymz} + \beta_3 C_{iymz} + \beta_4 E_{iymz} \times C_{iymz} \\ + \mu_m + Year + \zeta_z + \varepsilon_{iymz}, \qquad (3)$$

where $\ln R_{iymz}$ was the natural logarithm of the 24-h maximum rainfall that the woman experienced (or would have experienced) at her residential address, $E_{iymz}$ was the treatment binary variable and $y_{iymz}$ was the birth outcome. The variable $C_{iymz}$ was the number of 6-h advisories for which a woman's residence was within Hurricane Irene's cone of uncertainty. Such variation in advisories revealed the intensity of type I errors for those locations that experienced only mild weather exposures. The coefficient $\beta_4$ was the marginal effect of an additional advisory at the residence of an individual in the treatment group. While no advisories occurred for women in the control group, we calculated these advisories using the same approach to ensure that geographic factors were fully controlled. This approach, and interacting our advisory variable with our exposure indicator, ensured that our estimated marginal effects were unique to

exposed women, not driven by other geographic factors local to where advisories were issued.

## Reporting summary
Further information on research design is available in the Nature Portfolio Reporting Summary linked to this article.

## Data availability
The publicly available Parameter-elevation Regressions on Independent Slopes Model (PRISM) data and the American Community Survey (ACS) data used in this work can be accessed online at https://www.nhc.noaa.gov/gis/archive_forecast.php?year=2011 and https://www.census.gov/data/developers/data-sets/acs-5year, correspondingly. The vital statistics data obtained from the North Carolina Department of Health and Human Services (NCDHHS) contain sensitive information with individual-level health identifiers, which are protected and are not available due to data privacy laws. An anonymized data set for replication will be released in accordance with our NCDHHS data use agreement and upon request (send requests to JHochard@u-wyo.edu). Source Data are provided with this paper.

## Code availability
The code that supports the findings of this study is available upon request (send requests to JHochard@uwyo.edu).

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

## Acknowledgements
This project benefited from funding under U.S. Environmental Protection Agency awards RD836942, R840181 and National Science Foundation awards 1940141 and 1902282.

## Author contributions
J.H. conceptualized the research idea while J.H., Y.L. and N.A contributed equally to designing and executing the statistical strategy and drafting the manuscript.

## Competing interests
The authors declare no competing interests.
