## [Peer Review File · Nature Communications]

Peer review file,

Reviewer comments, first round

Reviewer #1 (Remarks to the Author):

this ecological study addresses the relationship between forecast accuracy and human health impacts in hurricane-threatened populations by studying pregnancy outcome in North Carolina as a consequence of Hurricane Irene landfall. The study of forecast accuracy is novel and adds important information to the general preparation and warning systems currently in place for natural disasters. The methods are appropriate and the results support the initial hypothesis. What would be interesting to know is the characteristics of the sample under study. Age, ethnicity, place of birth are main factors that come to mind as possible determinants of a stress response to a natural disaster, and describing the population of pregnant women in North Carolina within such a demographic framework would help better understanding how generalizable the results are. While an attempt was made to address possible water contamination as a factor playing a role in pregnancy outcomes, we don't have any information on the social support present around these women, and that could play a role in the observed outcomes.

Emanuela Taioli, MD PhD

Reviewer #2 (Remarks to the Author):

The authors introduce a very important and creative research question in the Abstract – what are the consequences of storm forecast accuracy on the health of pregnant women and their offspring? The authors compiled an extremely valuable and labor-intensive dataset in which to explore this question, consisting of 6 years of birth data and over 700,000 births.

However, the introduction does not set up how the study was conducted and how the results were obtained. That is, while the typical formatting of this journal provides methodological details after the discussion, there is a complete disconnect in this paper from the third paragraph of the Introduction into the Results section. There must be some general description of the study and how the research question will be addressed. That is, before the results are presented, the reader must be told how the hypothesized outcomes will be examined in a real-world study.

In addition, there should be clear hypotheses about both questions – first, what is the impact of storm exposure on birth outcomes?, and second, what is the impact of inaccurate storm exposure on birth outcomes? Those appear to be two entirely separate – and equally important – questions. In fact, I needed to skip from the introduction to page 11 (Data and Methods) before I had any idea what the investigators had done to explain the Results section.

Results:

1) The first sentence of the Results section reports: “We find that in utero exposure to Hurricane Irene created widespread and detrimental impacts to birth outcomes.” It seems to me that such a statement would require a comparison between birth outcomes during the hurricane (treatment

group) vs. the 4-5 years beforehand (control group). Where do such results appear?

2) The first set of results appears to report the overall effects of storm exposure on birth outcomes (pp. 3-6). The second set of results (starting on page 7) appears to report the effects of “hurricane anticipation” and storm forecast accuracy. However, most of the Introduction addresses the second set of results and inadequately sets up the first set. Moreover, all of the Discussion section is focused on the second set of results, without adequate attention to the implications of the first set of findings.

3) I am confused by the sentence that appears in the first paragraph of the Results “Importantly, we find that birth impacts *do not vary meaningfully* across storm exposure intensity...”, but then the authors go on to report (in several paragraphs and Figure 1 on page 4) the range of values on six birth outcomes between hurricane force winds/rain vs. mild rain/winds. Does this mean that there are no statistical differences between the endpoints or that there are no clinical differences in outcomes? This is further complicated by the sentence on page 5 that reads: “Each of these birth impacts are significant but statistically indistinguishable in magnitude across our wind and rainfall indicators for exposure intensity.” What does that mean – are they significant? Or are they statistically indistinguishable? These phrases are confusing and contradictory, at best.

4) The authors assert in the Results section that “Reinforcing our suspicion that observed birth impacts are driven by a mechanism other than the physical impacts of Hurricane Irene... (p. 6)”, but then examine a link between storm intensity and private well contamination rates, the connection which is unclear to me.

5) There is a great deal of speculation in the Results section, which seems misplaced. The Results should be a report of the statistical analyses conducted; it should not speculate on the meaning of such analyses. Therefore, sentences such as “Such a finding suggests that the anticipation of hurricane exposures and associated institutional responses to that anticipation, rather than the physical impacts from the storm itself, may be the driving force that disrupts healthcare services (p. 7)” do not belong in a Results section.

6) Table 1, which is presumably reporting results of some analysis, is not referenced or explained in the Results section at all and I do not understand exactly what data are being reported. Table 1 is mentioned briefly on page 14, but without explanation of what is being reported and what the numbers in the table mean.

Discussion

What are the *clinical* implications of the birth impacts reported in this paper? Significance levels are rarely reported. Apparently, the largest treatment effect was 14.4 g (discussed on p. 3), but the reader is never told whether the differences described across groups are clinically meaningful. Therefore, it is extremely difficult for the reader to ascertain the importance of the results reported.

Data and Methods:

1) Note that this sentence on page 11 is inferred, rather than demonstrated: “We focus on the effects of in utero exposure to disaster stress by identifying women who *anticipated direct hurricane impact but were not necessarily exposed* to severe

weather because of the storm's changing trajectory.”

Because women's anticipation of hurricane exposure was not assessed directly, the authors should be very careful about making such assertions.

Minor points:

1. Many paragraphs in the results are written in present tense (e.g., results that report well-water contamination, starting on page 5). In fact, results switch back and forth from past to present tense. I believe that it is most appropriate to present the Results section in past tense (since the analyses have already been conducted and are now being reported).
2. Why are the Data and Methods sections also described in present tense?
3. References need a careful review for capitalization, among other errors.

Reviewer #3 (Remarks to the Author):

Summary:

This is an interesting manuscript describing an examination of the effect of Hurricane Irene on birth outcomes using a large population-based sample of pregnant women in North Carolina. The primary focus is to evaluate the impact of forecast timing and accuracy on birth outcomes, but the authors also assessed potential water contaminants, geography, SES, and other risk factors. The authors found that exposure to Hurricane Irene was associated with reduced birth weight, gestational age, and other adverse birth outcomes, but the effect did not differ across intensities of hurricane exposure. They also reported that increased forecast advisories were associated with lower risk of adverse birth outcomes within the heavily exposed, but marginally higher risk within the least exposed.

Concerns:

1. The authors conclude that early hurricane advisories that over predicted the severity of the event caused a net increase in adverse birth outcomes among this population by postponing prenatal care in areas of low hurricane exposure. However, this is an overstatement of their actual findings. The authors show that the exposed group had fewer prenatal appointments compared to the un-exposed (pre-Irene) group, but they do not show any relationship between the number of appointments and birth outcomes. It is possible that women with healthier pregnancies may actually require fewer appointments, and the impact of missed prenatal care may be much different for a woman in the late third trimester compared to a woman in her first trimester. In light of this, the title of the manuscript should also be changed.
2. The authors measured water contaminants in samples collected from the area and found that the number of samples that exceeded EPA limits did not vary with storm intensity. From this, the authors further conclude that the association between hurricane exposure and adverse birth outcomes is not due to physical effects. Again, the authors have no real basis for this conclusion.

Some of the measured contaminants can impact birth outcomes below the EPA limit, and storm severity may actually be associated with differences in contamination, but at levels also below the EPA limit. Using the continuous concentrations (or more categories, e.g. quartiles) would be much more useful compared to the dichotomous variable.

3. If earlier hurricane warnings are truly associated with higher risk of adverse birth outcomes among women who are ultimately not highly exposed, stress could also be a potential mediator. This should be discussed as a possibility, rather than assuming the increased risk is related to slight changes in prenatal care.

4. Does Figure 3 show the location of all exposed women in the study? If so, the data is very sparse from the area with the highest intensity exposure. Did the authors consider how this might impact their analyses? Is it possible that women who were about to give birth or had high-risk pregnancies left the area because they were in direct line of the hurricane, and therefore were not in the study? This has happened in other disaster situations and studies, and I am unsure if the authors would have been able to capture these women. What does the distribution of births over time look like? At the minimum, the authors should acknowledge this limitation of their study.

5. More information in the methods section on the statistical analysis would be helpful for readers.

6. I am concerned about the authors' conclusion that delaying advisories might be better for pregnancy outcomes. This might be true in this one example, but further study is certainly needed before this is put into practice. Otherwise, pregnant women could be at higher risk if they do not have adequate time to prepare. The authors should state this in the discussion.

Responses to Referees

We are grateful for the opportunity to conduct a major revision of our manuscript NCOMMS-21-16795, which we have now completed. The reviewers' comments were supportive and instructive, and we feel that we have addressed them all thoroughly. As a result, the re-submitted manuscript is much improved.

Among other changes, a few major revisions include the following: 1) a revised paper title, 2) integration of socioeconomic and demographic data to gauge the sample's representativeness, 3) an expanded groundwater analysis 4) heavily revised organization of the manuscript and refined interpretation of key findings to avoid over concluding, and 5) additional literature references. We have also enlisted the help of an expert in the field, Dr. Nino Abashdize, who supported the manuscript's heavy revision. She has been instrumental in our efforts and now appears as a third author on the revised manuscript.

Reviewer #1 (Remarks to the Author):

This ecological study addresses the relationship between forecast accuracy and human health impacts in hurricane-threatened populations by studying pregnancy outcome in North Carolina as a consequence of Hurricane Irene landfall. The study of forecast accuracy is novel and adds important information to the general preparation and warning systems currently in place for natural disasters. The methods are appropriate and the results support the initial hypothesis.

What would be interesting to know is the characteristics of the sample under study. Age, ethnicity, place of birth are main factors that come to mind as possible determinants of a stress response to a natural disaster, and describing the population of pregnant women in North Carolina within such a demographic framework would help better understanding how generalizable the results are. While an attempt was made to address possible water contamination as a factor playing a role in pregnancy outcomes, we don't have any information on the social support present around these women, and that could play a role in the observed outcomes.

Emanuela Taioli, MD PhD

Thank you for your review and helpful comments, Dr. Taioli. We agree that a better understanding of sample demographics would help assess the extent to which our findings are generalizable to other hurricane-threatened communities. Our statistical approach takes great efforts to "control" these factors to prevent their confounding of our results. However, as you point out, the relevance of findings is limited to our sample and may not generalize if that sample is particularly unique. We have linked zip code-level US Census Data with (i) our existing sample dataset (baseline and treatment groups) and (ii) a larger zip code-level historical record of North Carolina births. To examine generalizability, we followed your suggestion to inspect age (2011 ACS Census Survey) and race indicators (2010 Decennial Census Survey) and included an income indicator (2011 ACS Census Survey). We do not have access to data on where women in our sample were born or their previous residential addresses.

Owing to the large scale of our analysis, we find that our sample includes 98.7% of all North Carolina births occurring between 8/26/2006 and 6/4/2012, which represents well the state’s relevant population. Within the sample, approximately 12.7% were “treated” births and 87.3% were “control” births with similar spatial distributions across the state. We compare 2010 and 2011 zip code-level census indicators for age, race and income weighted by the spatial distribution of births in the treated, control and omitted groups. We add a fourth group for comparison that weighs each of these indicators by the zip code-level distribution of North Carolina births from 1/1/1996 to 8/25/2006 and 6/5/2012 to 12/31/2017. This fourth comparison represents a spatial distribution based on 2.5 million out-of-sample births that occurred before and after our paper’s analysis.

We show that the magnitudes of these indicators are similar for any reasonable comparison

across our four groups suggesting that our sample is generalizable to at least the North Carolina population. These comparisons are now included in the paper’s appendix and referenced within the main text. We would expect the statewide representative sample would also mirror other non-North Carolina hurricane-threatened communities, but we do not have access to the required data to fully validate that extension here.

Reviewer #2 (Remarks to the Author):

The authors introduce a very important and creative research question in the Abstract – what are the consequences of storm forecast accuracy on the health of pregnant women and their offspring? The authors compiled an extremely valuable and labor-intensive dataset in which to explore this question, consisting of 6 years of birth data and over 700,000 births.

However, the introduction does not set up how the study was conducted and how the results were obtained. That is, while the typical formatting of this journal provides methodological details after the discussion, there is a complete disconnect in this paper from the third paragraph of the Introduction into the Results section. There must be some general description of the study and how the research question will be addressed. That is, before the results are presented, the reader must be told how the hypothesized outcomes will be examined in a real-world study.

In addition, there should be clear hypotheses about both questions – first, what is the impact of storm exposure on birth outcomes?, and second, what is the impact of inaccurate storm exposure on birth outcomes? Those appear to be two entirely separate – and equally important – questions.

In fact, I needed to skip from the introduction to page 11 (Data and Methods) before I had any idea what the investigators had done to explain the Results section.

We appreciate the reviewer pointing out this rough transition and agree that the exposition could be improved to adequately set up the “Results” section. As encouraged, we have added a fourth introduction paragraph that introduces the two primary research questions with corresponding hypotheses. We also offer enough insight into our experimental design/empirical approach that the reader has sufficient context to understand the ensuing “Results” section without skipping disruptively forward to the “Data and Methods” section. The new paragraph is copied here for the reviewer’s convenience:

“The purpose of this study is twofold. First, we investigate empirically the impact of in utero exposure to Hurricane Irene on a variety of birth outcomes, including birth weight, gestation length and incidence of low birth weight and preterm birth outcomes. The sample focuses on nearly 700,000 births in representative communities across North Carolina (Fig. A.1). Consistent with previous disaster and stressful events literature, we hypothesize that the hurricane exposure will reduce birth weights and gestation periods leading to more frequent low birth weight and preterm birth outcomes. Examination of Hurricane Irene as a natural experiment is conducted where birth outcomes leading up to the storm’s landfall serve as a baseline of comparison against births within the same zip code that occurred after the storm and experienced in utero exposures. Second, we investigate potential mechanisms underlying observed birth impacts with the expectation that groundwater contamination and intensive rainfall and wind would be an important contributor to measured birth outcome effects. Instead, we find that physical exposures alone (e.g., intensive rainfall and wind, resulting groundwater contamination) do not appear to explain observed effects. Rather, disaster anticipation leads to delayed and cancelled prenatal care, which, alongside other direct impacts of anticipation, such as the physiological impacts of stress and public services disruption (e.g., emergency medical care, nutrition access, etc.), may contribute to storm-impacted birth outcomes. Findings suggest that increasing storm forecast accuracy may promote healthy birth outcomes in regions that are threatened by uncertain storm events.”

Results:

1) The first sentence of the Results section reports: “We find that in utero exposure to Hurricane Irene created widespread and detrimental impacts to birth outcomes.” It seems to me that such a statement would require a comparison between birth outcomes during the hurricane (treatment group) vs. the 4-5 years beforehand (control group). Where do such results appear?

These results are shown in Figure 1 that we now reference following this sentence. These “treatment effects” are presented in difference from our baseline group. We expect that the new fourth Introduction paragraph, which offers a glimpse into our empirical strategy, better motivates how to interpret these effects against their baseline of comparison. To further reinforce this explanation, we added the following “Treatment effects are estimated against a baseline of births from the same zip code that occurred with expected delivery dates within the five years leading up to the Hurricane Irene’s disaster declaration date of August 25th, 2011.” to the Figure 1 caption.

2) The first set of results appears to report the overall effects of storm exposure on birth outcomes (pp. 3-6). The second set of results (starting on page 7) appears to report the effects of “hurricane anticipation” and storm forecast accuracy. However, most of the Introduction addresses the second set of results and inadequately sets up the first set. Moreover, all of the Discussion section is focused on the second set of results, without adequate attention to the implications of the first set of findings.

The reviewer is correct that we chose to motivate and emphasize the second set of results rather than the first. We view the second set of results as the particularly novel piece of this work whereas the disruptive impact of tropical cyclones (Currie et al. 2013) and other disaster events (Almond and Currie 2011; Aizer and Currie 2014; Black et al. 2016; Persson and Rossin-Slater 2018) on birth outcomes has been evidenced in the past. The first set of results do offer one of the first replications of Currie et al. 2013 (and stronger empirical evidence in a new study area) and are critical to establishing the motivation for the second set of results, which is important to emphasize. We now reference the Currie et al. 2013 paper in our third Introduction paragraph. The new segue paragraph at the end of the Introduction also motivates the importance of the two-part analysis as necessary to drawing our primary conclusions. We have added two new lead sentences in the Discussion

“The evidence presented in this work is consistent with the notion that in utero tropical storm exposures create abnormal birth conditions (46). Consistent with the broader disasters literature (9–12), we build on (46) with clear evidence that storm exposures reduce birth weights and gestation lengths while increasing the likelihood of preterm and low birth weight outcomes.”

emphasizing this literature and the first part of the contribution. Because the second set of findings is the pathway to a viable policy response, we continue to emphasize these findings throughout the remainder of the discussion.

3) I am confused by the sentence that appears in the first paragraph of the Results “Importantly, we find that birth

impacts *do not vary meaningfully* (italics added) across storm exposure intensity...”, but then the authors go on to report (in several paragraphs and Figure 1 on page 4) the range of values on six birth outcomes between hurricane force winds/rain vs. mild rain/winds. Does this mean that there are no statistical differences between the endpoints or that there are no clinical differences in outcomes? This is further complicated by the sentence on page 5 that reads: “Each of these birth impacts are significant but statistically indistinguishable in magnitude across our wind and rainfall indicators for exposure intensity.” What does that mean – are they significant? Or are they statistically indistinguishable? These phrases are confusing and contradictory, at best.

Thank you for pointing out this potential point of confusion. We have modified our language in the first paragraph of the results section to clarify when we are discussing (i) a treatment effect that has statistical significance in difference from zero and (ii) “statistically indistinguishable” as a statistical comparison between treatment effects (rather than in difference from zero).

“Herein we reported birth impacts as an average across all rainfall intensity bands with associated 95% confidence intervals (CI) in parentheses. Across these exposure intensities (Fig. 1), we detected consistent birth effects that were statistically distinguishable from zero. Importantly, the magnitudes of these estimated effects did not increase with storm exposure intensity - i.e., birth impacts that were estimated for individuals who experienced intensive rainfall and hurricane force winds were not statistically distinguishable from birth effects among individuals who experienced modest wind and rainfall intensities.”

Further, the second sentence referenced by the referee that appeared on page 5 now appears at the bottom of page 4 and has been rewritten for clarity as:

“Each of these birth impacts were statistically significant in difference from zero but the magnitudes of these effects were statistically indistinguishable from each other across our wind and rainfall indicators for exposure intensity (Fig. 1).”

4) The authors assert in the Results section that “Reinforcing our suspicion that observed birth impacts are driven by a mechanism other than the physical impacts of Hurricane Irene... (p. 6)”, but then examine a link between storm intensity and private well contamination rates, the connection which is unclear to me.

Here, we are referencing groundwater contamination as one of the “physical effects” of the storm, which we now understand may be confusing. Rather, direct storm effects such as high winds and flooding are more aptly referred to as “physical effects”. Referring to outcomes like groundwater contamination that originate from storm-related flooding (e.g., second-order effects) has the potential to create confusion. This sentence has been rewritten as

“Reinforcing our suspicion that observed birth impacts are driven by a mechanism other than the direct physical impacts of Hurricane Irene (e.g., high winds or flooding) or indirect physical exposures (e.g., groundwater contamination that resulted in flooded areas), we found no meaningful relationship between storm exposure intensity and private well water contamination rates (Fig. A.1).”

In addition, there might be confusion about why we are presenting null results related to groundwater contamination in the first place. The purpose of presenting these null results in Fig. A.1 is to support the paper’s broader body of empirical evidence that the storm’s physical impacts, whether measured directly using rainfall and wind intensities or indirectly from groundwater contamination rates, are unlikely to be the mechanism driving the observed treatment effect on birth outcomes.

5) There is a great deal of speculation in the Results section, which seems misplaced. The Results should be a report of the statistical analyses conducted; it should not speculate on the meaning of such analyses. Therefore, sentences such as “Such a finding suggests that the anticipation of hurricane exposures and associated institutional responses to that anticipation, rather than the physical impacts from the storm itself, may be the driving force that disrupts healthcare services (p. 7)” do not belong in a Results section.

We have addressed this comment by moving and editing some of the Results section into the Discussion section. Specifically, we combined the following sentences from pages 4, 6 and 7 and moved them into the Discussion section:

“We observed evidence that in utero exposure to Hurricane Irene created widespread and detrimental impacts to birth outcomes.”

“Across the exposure intensities (Fig. 1) we detected consistent birth effects that were statistically distinguishable from zero. Importantly, the magnitudes of these estimated effects did not increase with storm exposure intensity - i.e., birth impacts that were estimated for individuals who experienced intensive rainfall and hurricane force winds were not statistically distinguishable from birth effects among individuals who experienced modest wind and rainfall intensities.”

“We would expect that a higher intensity of rainfall and wind would be associated with more drastic birth impacts. The consistency in our measured birth impacts across storm exposure intensity was unexpected and may suggest that birth impacts were being driven by a mechanism other than physical storm exposures (e.g., rainfall, wind and groundwater contamination in flooded areas).”

“During Hurricane Irene, over 2 million individuals that represent over 20% of North Carolina’s population (24) relied on private wells that are federally unregulated and particularly vulnerable to contamination from severe weather and flooding events (25–28).”

“This approach provided an indicator of potential environmental exposures that may explain the observed statewide hurricane-linked birth impacts. Reinforcing our suspicion that observed birth impacts are driven by a mechanism other than the direct physical impacts of Hurricane Irene (e.g., high winds or flooding) or indirect physical exposures (e.g., groundwater contamination that resulted in flooded areas).”

“Systematic geographic sorting along socioeconomic lines, which may have occurred during our sample window, could reasonably explain observed birth impacts. In such a case, we would expect prediagnosed medical risk factors to vary systematically between our treatment and control groups.”

“Both prenatal care indicators suggested that hurricane exposure creates a disruption of access to healthcare services (Fig. 2).”

“Such a finding suggests that the anticipation of hurricane exposures and associated institutional responses to that anticipation, rather than the direct or indirect physical impacts from the storm itself, may be the driving force that disrupts healthcare services.”

We have combined these sentences and rewritten as (see Discussion section, pages 10 and 11):

“We observed evidence that in utero exposure to Hurricane Irene created widespread and detrimental impacts to birth outcomes. Across all rainfall intensity bands analyzed in the paper, we detected consistent birth effects that were statistically distinguishable from zero. Although we would expect that a higher intensity of rainfall and wind would be associated with more drastic birth impacts, the magnitudes of these estimated effects did not increase with storm exposure intensity. The consistency in our measured birth impacts across storm exposure intensity was unexpected and may suggest that birth impacts were being driven by a mechanism other than physical storm exposures.”

“We might expect that statewide groundwater contamination may explain the observed statewide hurricane-linked birth impacts. During Hurricane Irene, over 2 million individuals that represent over 20% of North Carolina’s population (24) relied on private wells that were federally unregulated and particularly vulnerable to contamination from severe weather and flooding events (25-28). We found no evidence that observed birth impacts were driven by a mechanism other than the direct physical impacts of Hurricane Irene (e.g., high winds or flooding) or indirect physical exposures (e.g., groundwater contamination that resulted in flooded areas).”

“We further found no evidence that the observed birth impacts were explained by a systematic geographic sorting along socioeconomic lines, which may have occurred during our sample window. In such a case, we would expect prediagnosed medical risk factors to vary systematically between our treatment and control groups.”

“Importantly, we observed evidence that hurricane exposure created prenatal care disruptions that varied little across the intensity of storm exposures. Such a finding suggests that the anticipation of hurricane exposures and associated institutional responses to that anticipation, rather than the physical impacts from the storm itself, may be the driving force that disrupts healthcare services.”

6) Table 1, which is presumably reporting results of some analysis, is not referenced or explained in the Results section at all and I do not understand exactly what data are being reported. Table 1 is mentioned briefly on page 14, but without explanation of what is being reported and what the numbers in the table mean.

Thank you for pointing out that we failed to reference this table’s results where appropriate. We now reference Table 1 in the final two paragraphs of the Results section. We have also moved the preceding three paragraphs to the Data and Methods section as they focus primarily on why we overlaid and examined the time spent within the National Hurricane Center’s Cone of Uncertainty on these three subpopulations (Rain>2in, Rain 1-2in and Rain ≤ 1in).

Discussion

What are the *clinical* implications of the birth impacts reported in this paper? Significance levels are rarely reported. Apparently, the largest treatment effect was 14.4 g (discussed on p. 3), but the reader is never told whether the differences described across groups are clinically meaningful. Therefore, it is extremely difficult for the reader to ascertain the importance of the results reported.

This is an excellent point. Indeed, in our submitted manuscript we focused only on the statistical relevance of these measured effects without adequate interpretation of their clinical relevance. We have now included a discussion paragraph at the bottom of page 10 interpreting these magnitudes from a clinical effects perspective:

The presented findings are clinically relevant in addition to being statistically robust. Relying on British National Child Development Survey data, (38) show that low-birth weight children weighing less than 2,500g were more than 25 percent less likely to pass high school English and math exit examinations and were also less likely to be employed at the age of 33. In our work, we show that in utero exposure to Hurricane Irene created a 2.52% to 10.34% increase in the likelihood of crossing this critical low birth weight threshold, which is increased further by storm exposure anticipation. For our birth weight outcomes, we find general effects that range from 0.17% to 0.61% and increase similarly with storm exposure anticipation. For the average individual in our lightly exposed sample (Rain ≤1in), direct effects of storm exposure and indirect effects from storm forecast-driven anticipation (on average 6.5 6-hour windows within the Cone of Uncertainty), cumulative birth weight reductions are approximately 1% to 2%. The clinical impacts of these birth weight reductions are uncertain. However, the measured magnitude is well below the commonly cited “10% change” where disruption to later life outcomes, such as high school graduation rates, IQ, income and height have been documented (39).

Data and Methods:

1) Note that this sentence on page 11 is inferred, rather than demonstrated:

*“We focus on the effects of in utero exposure to disaster stress by identifying women who *anticipated direct hurricane impact but were not necessarily exposed* (italics added) to severe weather because of the storm’s changing trajectory.”*

Because women’s anticipation of hurricane exposure was not assessed directly, the authors should be very careful about making such assertions.

We agree with the reviewer and have added “were likely” here to reduce the strength of this claim as hurricane anticipation was not measured directly.

We focused on the effects of in utero exposure to disaster stress by identifying women who were likely to anticipate direct hurricane impact but were not necessarily exposed to severe weather because of the storm’s changing trajectory.

Minor points:

1. Many paragraphs in the results are written in present tense (e.g., results that report well-water contamination, starting on page 5). In fact, results switch back and forth from past to present tense. I believe that it is most appropriate to present the Results section in past tense (since the analyses have already been conducted and are now being reported).

Thank you for pointing out this inconsistency in tenses. The Results section is now consistently in the past tense.

2. Why are the Data and Methods sections also described in present tense?

We have made the recommended change. The Data and Methods sections are now consistently in the past tense.

3. References need a careful review for capitalization, among other errors.

We have carefully reviewed the reference list and made corrections. Thank you for this point.

Reviewer #3 (Remarks to the Author):

Summary:

This is an interesting manuscript describing an examination of the effect of Hurricane Irene on birth outcomes using a large population-based sample of pregnant women in North Carolina. The primary focus is to evaluate the impact of forecast timing and accuracy on birth outcomes, but the authors also assessed potential water contaminants, geography, SES, and other risk factors. The authors found that exposure to Hurricane Irene was associated with reduced birth weight, gestational age, and other adverse birth outcomes, but the effect did not differ across intensities of hurricane exposure. They also reported that increased forecast advisories were associated with lower risk of adverse birth outcomes within the heavily exposed, but marginally higher risk within the least exposed.

Concerns:

1. The authors conclude that early hurricane advisories that over predicted the severity of the event caused a net increase in adverse birth outcomes among this population by postponing prenatal care in areas of low hurricane exposure. However, this is an overstatement of their actual findings. The authors show that the exposed group had fewer prenatal appointments compared to the un-exposed (pre-Irene) group, but they do not show any relationship between the number of appointments and birth outcomes. It is possible that women with healthier pregnancies may actually require fewer appointments, and the impact of missed prenatal care may be much different for a woman in the late third trimester compared to a woman in her first trimester. In light of this, the title of the manuscript should also be changed.

We agree with the reviewer and recognize now that we incorrectly lead the reader to believe a measured link between disrupted prenatal care and observed impacts on birth outcomes. Indeed, we never estimate this link between prenatal care indicators and birth outcomes... only document that the former are disrupted, and the latter are impaired. We do hope to conduct that analysis in the future.

To avoid over concluding, we have changed the paper title to:

Expecting Mother Nature: The Disruptive Impacts of Hurricanes and their Forecasts on Birth Outcomes and Prenatal Care

We also edited the following abstract sentence from

“Disaster anticipation disrupted healthcare services by delaying and canceling prenatal care leading to impaired birth outcomes.”

to

“Disaster anticipation disrupted healthcare services by delaying and canceling prenatal care.

We retained the following sentence in the final Introduction paragraph (but softened the language by replacing “are likely to” with “may”)

“Rather, disaster anticipation leads to delayed and cancelled prenatal care, which, alongside other direct impacts of anticipation such as stress and public services disruption (e.g., emergency medical care, nutrition access, etc.) may contribute to storm-impacted birth outcomes.”

We also edited to the following second-to-last Discussion paragraph

“In the case of Hurricane Irene, the early release of the storm track forecast triggered a precautionary response by patients and healthcare providers. The decision to cancel healthcare appointments was driven by risk averse preferences among these groups. As such, we discover that this combination of risk averse preferences and forecast uncertainty during Hurricane Irene disproportionately harmed the unborn.”

to

“In the case of Hurricane Irene, the early release of the storm track forecast is likely to have triggered a precautionary response by patients and healthcare providers. The apparent decision to cancel healthcare appointments may be driven by risk averse preferences among these groups. As such, it appears that this combination of risk averse preferences and forecast uncertainty during Hurricane Irene may have disproportionately harmed the unborn.”

Finally, we edited a sentence in the concluding paragraph in the following way:

“Trimester of exposure should also be investigated to identify populations that are most vulnerable to disruptions in health care services.

to

“Trimester of exposure should also be investigated to identify populations that are most vulnerable to disruptions in health care services and to formally validate whether prenatal care disruptions were the force driving observed birth outcomes.”

2. The authors measured water contaminants in samples collected from the area and found that the number of samples that exceeded EPA limits did not vary with storm intensity. From this, the authors further conclude that the association between hurricane exposure and adverse birth outcomes is not due to physical effects. Again, the authors have no real basis for this conclusion. Some of the measured contaminants can impact birth outcomes below the EPA limit, and storm severity may actually be associated with differences in contamination, but at levels also below the EPA limit. Using the continuous concentrations (or more categories, e.g. quartiles) would be much more useful compared to the dichotomous variable.

Indeed, there is evidence that *in utero* exposures below the established EPA limits. We reran our groundwater analysis using the most sensitive indicator of contamination (the North Carolina State Laboratory Detection Limits) that are between ½ to 1 order of magnitude less than the EPA limits. We replicate our finding that fails to detect any spikes in arsenic, cadmium, chromium, lead, manganese, and nitrate among exposed wells compared to their pre-storm and same zip code counterparts.

Inorganic	EPA level	NC State Laboratory Detection Limits
Arsenic	0.01	0.005
Cadmium	0.005	0.001
Chromium	0.10	0.01
Lead	0.015	0.005
Manganese	0.05	0.03
Nitrate	10	1.00

We do want to emphasize that the groundwater analysis presented is one part of a broader body of evidence that tells a cohesive empirical narrative. By no means does this analysis rule out the presence of environmental exposures from tropical storm events nor does it rule out that such exposures could be harmful to *in utero* development. These new groundwater results adopting the alternative threshold are now also reported in the paper's appendix. Given the scope of our dataset, however, we would like to investigate more formally the impact of hurricane exposures on groundwater contamination... especially near known surface sources of contamination like coal ash pits, intensive agriculture operations, etc. The authors are working on this analysis a part of a larger project and we hope to make that contribution soon in a distinct contribution.

In the final paragraph of the Introduction, we have softened this conclusion by editing

“Instead, we find that physical exposures alone (e.g., intensive rainfall and wind and resulting groundwater contamination) cannot explain observed effects.”

to

“Instead, we find that physical exposures alone (e.g., intensive rainfall and wind and resulting groundwater contamination) do not appear to explain observed effects.”

We also added the following sentence

“This null finding did not rule out contaminated groundwater exposures from Hurricane Irene and is not the primary purpose of our contribution. However, if these contamination events were the driving force behind our results, we might have expected such evidence to appear in this sample.”

at the end of the fourth results paragraph.

3. If earlier hurricane warnings are truly associated with higher risk of adverse birth outcomes among women who are ultimately not highly exposed, stress could also be a potential mediator. This should be discussed as a possibility, rather than assuming the increased risk is related to slight changes in prenatal care.

We have edited our text to avoid over concluding the role that prenatal care access may have in determining birth outcomes (referee #3 response #1 above). We do agree with the reviewer that physiological impacts of stress may be just as important (if not more important) than institutional responses that are responsive to a stressful situation. We do not want to minimize either of these channels or claim that we can fully disentangle their effects.

Here we try to be clear in our second-to-last Introduction sentence:

“Rather, disaster anticipation leads to delayed and cancelled prenatal care, which, alongside other direct impacts of anticipation such as the physiological impacts of stress and public services disruption (e.g., emergency medical care, nutrition access, etc.) may contribute to storm-impacted birth outcomes.”

Like the falsification tests provided for groundwater contamination. We do present evidence in the results that the exposed population of women had similar rates of gestational hypertension and eclampsia as the baseline population. Also like the groundwater contamination evidence, these tests alone are not sufficient to rule out maternal stress as the causal channel. Lastly, we edit the final discussion sentence to read:

“Here, future research should examine how public risk responses are likely to differ in disaster scenarios characterized by extreme ambiguity and the extent to which birth impacts from storm events are driven by physiological stress channels compared to institutional responses to situational stress.”

4. Does Figure 3 show the location of all exposed women in the study? If so, the data is very sparse from the area with the highest intensity exposure. Did the authors consider how this might impact their analyses? Is it possible that women who were about to give birth or had high-risk pregnancies left the area because they were in direct line of the hurricane, and therefore were not in the study? This has happened in other disaster situations and studies, and I am unsure if the authors would have been able to capture these women. What does the distribution of births over time look like? At the minimum, the authors should acknowledge this limitation of their study.

The reviewer makes a good point. Yes, Figure 3 does show all treated women within the study and those within the storm’s most certain path are going to be sparser than the full distribution of women across the state. It would also make sense that coastal pregnant populations – especially those with higher income – might be more mobile in response to an impending disaster threat. We would expect this resourced population to out-migrate to lower risk areas leaving the lower income (and higher risk as the reviewer notes) pregnant population in the storm’s track.

We are less concerned about hurricane-triggered outmigration to better hospitals during birth because our sample selects into treatment based on exposure during any part of the pregnancy term – i.e., *in utero exposure*. This is a very small portion of women who would be giving birth in a different hospital than their home community. This is relevant only for the share of women who were pregnant, residing in the most threatened communities and approaching the end of their term when the hurricane arrived.

Income-based outmigration may still affect our primary finding that Hurricane Irene impaired birth outcomes and secondary finding measuring the relative effect of earlier and uncertain storm forecasts. For the former, income-

based outmigration will drive our findings toward the null – i.e., we are less likely to detect an effect – because a portion of the population can avoid some of the most intense physical exposures. In the absence of this averting behavior, we might expect more severe birth impacts than those that we report.

In measuring the effectiveness of storm forecasts, income-based outmigration is one of the channels through which we would expect storm forecasts to operate – i.e., if you have an additional 6, 12, 18 or 24 hours of time to prepare, you have advanced warning to harden your home and/or evacuate the area. Indeed, we find some evidence that this advanced warning improves birth outcome for this severely impacted population and would expect that income-based outmigration is one of the underlying driving mechanisms (e.g., the ability to avoid exposure may reduce the stressfulness of the event and shelter *in utero* development and subsequent birth outcomes).

At the worst, income-based outmigration renders our findings and empirical strategy a conservative approach. This is a very interesting question deserving of further investigation. With our current data, the best we can do is argue that our dataset is broadly representative and that future deeper dives into heterogeneous treatment effects should be prioritized. For this reason and along a similar line of inquiry from Referee #1, we have integrated our dataset with US Census Data on income, race, and age to argue that our sample population is representative of omitted residents during the sample period and North Carolina residents before and after the sample period.

These comparisons are now included in the paper's appendix and referenced within the main text. We would expect the statewide representative sample would also mirror other non-North Carolina hurricane-threatened communities, but we do not have access to the required data to fully validate that extension here.

5. More information in the methods section on the statistical analysis would be helpful for readers.

We agree and follow the instruction of Referee #2 by adding a general introduction to the methodological approach at the end of the paper's first section. This introduction makes the transition from Introduction to Results much less harsh and easier to follow for the reader. In addition, we relocated three paragraphs of methods-related material that prior lived within the results section and integrated it into the Data and Methods section. The positioning of this material feels much more natural in the expanded methods section, which we think is much improved.

6. I am concerned about the authors' conclusion that delaying advisories might be better for pregnancy outcomes. This might be true in this one example, but further study is certainly needed before this is put into practice. Otherwise, pregnant women could be at higher risk if they do not have adequate time to prepare. The authors should state this in the discussion.

The authors agree that more study is needed before policy changes should be implemented from this paper's findings. We think the work is particularly novel as it is a unique finding that we hope will be replicated and further investigated in new contexts so that policymakers can evaluate the breadth of empirical support for potential policy options. We have revised the paper's concluding sentences to ensure that this disclaimer is as clear as possible:

“The ambiguity (rather than uncertainty) surrounding a low-information scenario may trigger similar precautionary responses by individuals and institutions during the anticipation phase of delayed official storm forecasts. Here, future research should examine how public risk responses are likely to differ in disaster scenarios characterized by extreme ambiguity and the extent to which birth impacts from storm events are driven by physiological stress channels compared to institutional responses to situational stress. Better understanding these mechanisms that underlie population responses and institutional responses to disasters is essential to guiding future policies that might affect disaster preparedness.”

We also add the following precautionary and transition sentence leading up to the Discussion's final paragraph

“Before such approaches could be used in a useful way, further study is needed to understand how disaster-threatened populations might respond to delayed but higher accuracy forecasts.”

Reviewer comments, second round

Reviewer #1 (Remarks to the Author):

The authors have addressed all the comments and have answered all questions in a satisfactory manner.

Reviewer #2 (Remarks to the Author):

The authors did an excellent job responding to the critiques from all 3 reviewers that were raised in the initial review. The paper is far improved.

I have one remaining recommendation:

While I appreciate the addition of the fourth paragraph of the introduction, which now sets up the study and the hypotheses, I would encourage the authors to delete the three sentences at the end of the paragraph that prematurely report the findings and recommendations. That is, I believe the following sentences, which appear at the top of page 4, should be deleted from the introduction: "Instead, we find that physical exposures alone (e.g., intensive rainfall and wind, resulting groundwater contamination) do not appear to explain observed effects. Rather, disaster anticipation leads to delayed and cancelled prenatal care, which, alongside other direct impacts of anticipation, such as the physiological impacts of stress and public services disruption (e.g., emergency medical care, nutrition access, etc.), may contribute to storm-impacted birth outcomes. Findings suggest that increasing storm forecast accuracy may promote healthy birth outcomes in regions that are threatened by uncertain storm events."

This text is more appropriately situated in the Discussion.

Beyond this issue, I have no further concerns.

Reviewer #3 (Remarks to the Author):

This manuscript describes an analysis of the effect of Hurricane Irene, including strength of exposure and accuracy of pre-storm warnings, on birth outcomes in North Carolina. The authors have addressed the majority of concerns from previous reviewers, but the manuscript still needs some clarifications.

Abstract:

I agree with a previous reviewer that the authors overstated their findings between over-prediction of severity and adverse birth outcomes. In their response, the authors state that the problematic sentence in the abstract had been changed, but it still reads "Disaster anticipation disrupted healthcare services by delaying and canceling prenatal care leading to impaired birth outcomes".

Figures:

The majority of figures show a relationship between hurricane severity (rainfall) and an outcome (e.g., birth weight) overlaid with the cumulative distribution of participants according to rainfall. Is there any relationship between the y-axis (outcome) and the cumulative distribution? The placement of the regression line within the cumulative distribution is confusing and not explained anywhere (that I could find). Also – cumulative is spelled incorrectly in all of the figures.

Table 1. There is no label for the middle row of numbers in parentheses for each outcome/exposure category. Please clarify.

Page 14-15: Please provide a more general explanation and interpretation of what exactly the equations listed are being used for. The equations and detailed methods are helpful, but some readers will appreciate a more basic description.

Table A.1: Mean and SD statistics for binary outcomes are not really helpful. These should be presented as frequencies.

Responses to Referees

Referees: we have now completed the second revision to our manuscript (NCOMMS-21-16795) and are grateful for the time that you have taken throughout this process. Our project team was delighted to hear that Reviewer #1 was satisfied with our first-round revisions. We also appreciated the feedback from Reviewer #2 and agree that the manuscript has improved drastically through the peer review process. Reviewer #2, we have addressed your one final suggestion in the manuscript as described below. Lastly, we appreciate the thoughtful additional round of feedback from Reviewer #3 and have revised the manuscript to address each of these constructive suggestions.

Reviewer #1 (Remarks to the Author):

The authors have addressed all the comments and have answered all questions in a satisfactory manner.

Reviewer #2 (Remarks to the Author):

The authors did an excellent job responding to the critiques from all 3 reviewers that were raised in the initial review. The paper is far improved.

I have one remaining recommendation:

While I appreciate the addition of the fourth paragraph of the introduction, which now sets up the study and the hypotheses, I would encourage the authors to delete the three sentences at the end of the paragraph that prematurely report the findings and recommendations. That is, I believe the following sentences, which appear at the top of page 4, should be deleted from the introduction: "Instead, we find that physical exposures alone (e.g., intensive rainfall and wind, resulting groundwater contamination) do not appear to explain observed effects. Rather, disaster anticipation leads to delayed and cancelled prenatal care, which, alongside other direct impacts of anticipation, such as the physiological impacts of stress and public services disruption (e.g., emergency medical care, nutrition access, etc.), may contribute to storm-impacted birth outcomes. Findings suggest that increasing storm forecast accuracy may promote healthy birth outcomes in regions that are threatened by uncertain storm events."

This text is more appropriately situated in the Discussion.

Beyond this issue, I have no further concerns.

Thank you, Reviewer #2, we have now deleted the three referenced sentences and have left this summarization of our results for the "Discussion" section.

Reviewer #3 (Remarks to the Author):

This manuscript describes an analysis of the effect of Hurricane Irene, including strength of exposure and accuracy of pre-storm warnings, on birth outcomes in North Carolina. The authors have addressed the majority of concerns from previous reviewers, but the manuscript still needs some clarifications.

Abstract:

I agree with a previous reviewer that the authors overstated their findings between over-prediction of severity and adverse birth outcomes. In their response, the authors state that the problematic sentence in the abstract had been changed, but it still reads "Disaster anticipation disrupted healthcare services by delaying and canceling prenatal care leading to impaired birth outcomes".

Thank you for drawing our attention to this language. Indeed, we present no evidence that disrupted access to healthcare services itself is a driving force behind observed birth impacts. To avoid over concluding, we have relaxed our language in the abstract:

"Disaster anticipation disrupted healthcare services by delaying and canceling prenatal care, which, alongside other direct impacts of anticipation, such as the physiological impacts of stress and public services disruption (e.g., emergency medical care, nutrition access, etc.), may contribute to storm-impacted birth outcomes."

Figures:

The majority of figures show a relationship between hurricane severity (rainfall) and an outcome (e.g., birth weight) overlaid with the cumulative distribution of participants according to rainfall. Is there any relationship between the y-axis (outcome) and the cumulative distribution? The placement of the regression line within the cumulative distribution is confusing and not explained anywhere (that I could find). Also – cumulative is spelled incorrectly in all of the figures.

There is no empirical relationship between the y-axis and the cumulative distribution. Because rainfall intensity is only a proxy for physical storm exposures, the estimated treatment effects are overlaid with a cumulative distribution of wind speed exposures to allow the reader to examine visually the association between birth outcomes and wind speed. For example, the largest treatment effect (14.4 g) was estimated for mothers experiencing hurricane winds and over 10 inches of rainfall, while the smallest treatment effect (10.1g) was estimated for mothers experiencing only mild winds and less than 1 inch of rainfall (Figure 1.a).

We added the following sentence to the caption of Figures 1, 2, A.2, A.3, and A.4:
“Estimated treatment effects were overlaid with a cumulative distribution of wind speed exposures.”

We also added the following explanation to the first paragraph of the results section (page 4):
“The estimated birth impacts were overlaid with a cumulative distribution of wind speed exposures in order to visually explore the association between estimated birth outcomes and wind intensity (Fig. 1).”

Thank you also for pointing out the repeated typo in each figure. These have now been corrected.

Table 1. There is no label for the middle row of numbers in parentheses for each outcome/exposure category. Please clarify.

We now include a note at the end of the table stating that the standard errors, clustered at the county level, are in parentheses. We also added information about the statistical significance of the coefficient estimates: $***P < 0.01$, $**P < 0.05$, and $*P < 0.1$.

Page 14-15: Please provide a more general explanation and interpretation of what exactly the equations listed are being used for. The equations and detailed methods are helpful, but some readers will appreciate a more basic description.

We added the following explanation to the Data and Methods section (p 14-15) to offer a more general explanation of our empirical approach:

“To measure the impact of hurricane exposures, we compared birth outcomes among two groups of women who lived in the same zip codes and experienced antenatal *or* postnatal exposures. The comparison groups included a “treatment” group of exposed women whose births may have been affected through *in utero* exposures to the physical impacts of Hurricane Irene and a “control” group of exposed women whose birth outcomes predated the hurricane’s arrival and could not have been impacted by its physical impacts. To refine our comparison groups, high-resolution rainfall intensities were predicted at each woman’s residential address and implemented as our proxy exposure variable. While unlikely within any given zip code, if Hurricane Irene’s physical exposures were systematically correlated with neighborhoods that were underserved in other ways that might impact birth outcomes (e.g., access to healthcare services or insurance), conditioning on Hurricane Irene’s rainfall intensities ensured that our treatment and control groups mirrored one another. More specifically, we estimated the following equation”

“We first estimated the average treatment effect of hurricane exposure on birth outcomes and then calculated the predicted birth outcomes for each woman in our sample based on their actual exposure. More specifically, after equation (1) was estimated, we calculated the predicted impact on an individual woman’s birth outcome, ψ , as a function of $R_{iy mz}$ ”

We also modified the following sentence on page 15 from:

“To examine the effect of hurricane anticipation on birth outcomes, we augmented the estimating equation as follows”

To:

“To examine the effect of hurricane anticipation (i.e., an additional six-hour window within Hurricane Irene’s predicted “cone of uncertainty”) on birth outcomes, we augmented the estimating equation as follows”

Table A.1: Mean and SD statistics for binary outcomes are not really helpful. These should be presented as frequencies.

We split Table A.1 into two Panels: A and B. We removed summary statistics of the binary outcome variables from Panel A and reported frequencies for binary outcomes in Panel B.

Reviewer comments, third round

Reviewer #3 (Remarks to the Author):

The authors have addressed my comments and concerns.